# Native Spatio-temporal 4D Variational AutoEncoder

**Lihe Ding** [1 2]   **Weicai Ye** [† 2]   **Shaocong Dong** [3]   **Xintao Wang** [2]   **Pengfei Wan** [2]   **Kun Gai** [2]   **Tianfan Xue** [† 1 4]

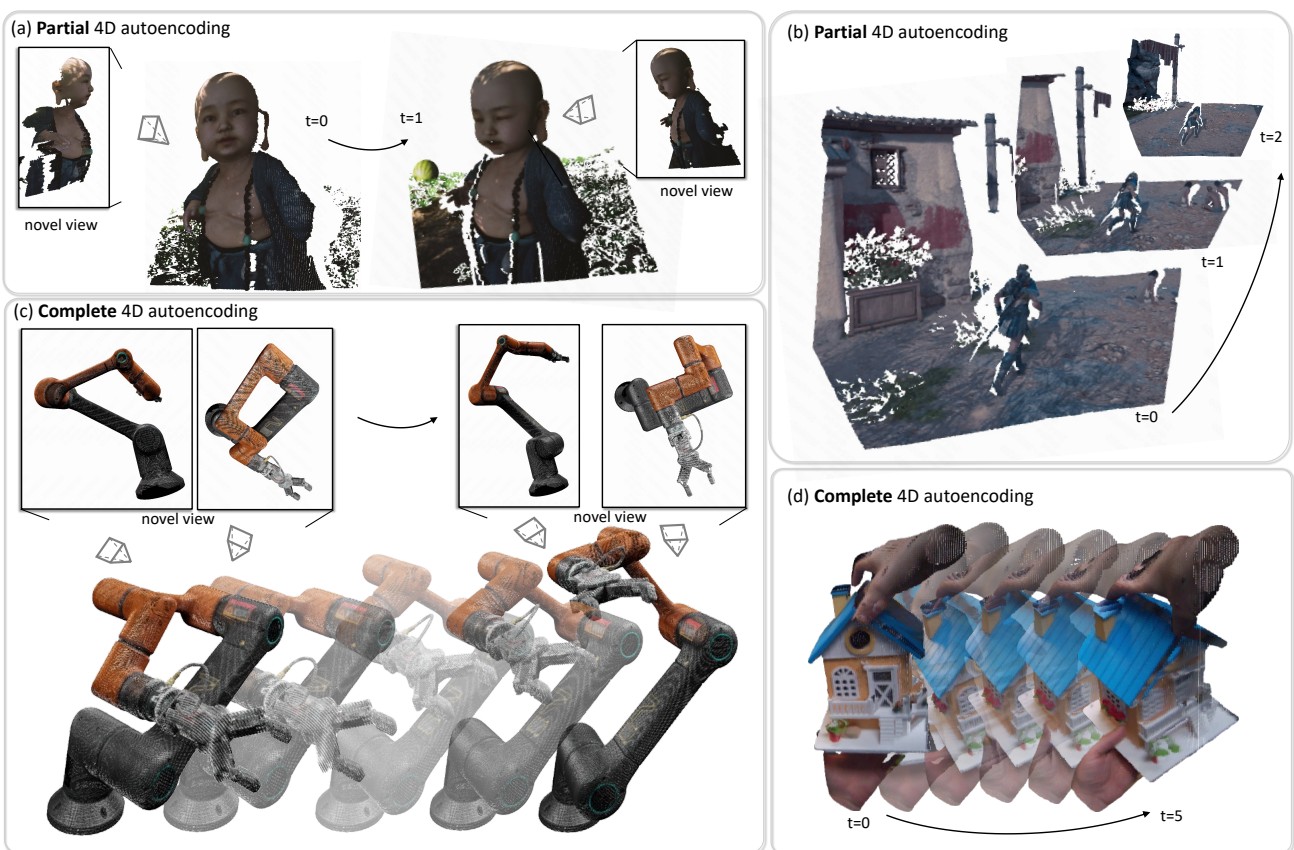

*Figure 1.* Our native spatio-temporal 4D VAE achieves high-quality reconstruction for both partial and complete 4D content.

## Abstract

Dynamic 3D content representation is crucial for generating moving 3D objects and scene. Existing 4D variational autoencoders (VAEs) are mainly based on projected 2D pointmaps, which are only incomplete and view-dependent observations that do not model the native 4D positional relations between points. This often leads to projection-induced distortions and irreversible token dislocation. In this paper, we introduce a novel 4D VAE that operates directly in native 4D space, that is dynamic colored voxel space, without 2D projection. This preserves explicit spatio-temporal coordinates throughout the learned encoder and decoder, enabling both partial and complete 4D content encoding. To support a flexible temporal compression ratio, we also design a novel spatio-temporal window attention module that performs attention within local 4D windows. Additionally, we propose a differentiable voxel rendering loss based on sparse voxel rasterization to improve the geometry

[1]MMLab, The Chinese University of Hong Kong, Hong Kong, China [2]Kuaishou Technology, China [3]Department of Computer Science and Engineering, Hong Kong University of Science and Technology, Hong Kong, China [4]CPII under InnoHK. Correspondence to: Weicai Ye <maikeyeweicai@gmail.com>, Tianfan Xue <tfxue@ie.cuhk.edu.hk>.

*Proceedings of the $43^{rd}$ International Conference on Machine Learning*, Seoul, South Korea. PMLR 306, 2026. Copyright 2026 by the author(s).

and color reconstruction quality. On 4D reconstruction tasks, our approach improves reconstruction fidelity over pointmap VAEs and flow-based VAEs while learning a more structurally consistent latent space. We further demonstrate the generative potential of our method by training a video-conditioned 4D diffusion model. Project website: https://native4d.github.io/.

## 1. Introduction

Dynamic 3D content generation and compression have become pivotal for emerging applications in virtual/augmented reality, autonomous driving, robotics, and digital twins. For both tasks, a powerful 4D latent presentation is critical, and these latents are learned through 4D variational autoencoders (VAEs). Existing 4D VAE mainly rely on pointmap-based architectures (Wang et al., 2024; Xu et al., 2025; Chen et al., 2025b; Jiang et al., 2025b), which shows a promising result in modeling partial observations (e.g., single-viewpoint scans). These works adapt the video diffusion model to the pixel-aligned pointmap latent for monocular partial 4D reconstruction.

However, these 2D pointmap-based VAEs suffer some critical issues when applying to complete and high-quality 4D content compression or reconstruction. First, 2D pointmap-based VAEs can only encode incomplete and view-dependent observations (2D projection of a 3D scene), and cannot model a full 4D scene geometry and appearance, like the moving arm showing Fig. 1(c). To extend them to full 4D, they either aggregate from arbitrary viewpoints or extracted from dynamic meshes, making them difficult for downstream training. Second, by projecting 3D points onto image planes with 3D coordinates as pixel values, these methods eliminate native 4D positional relationships between points. As results, they often introduce two types of artifacts: (i) projection-induced distortions (e.g., motion blur, stretching, and temporal flickering) due to the loss of continuous spatiotemporal coherence, and (ii) irreversible token dislocation, as 2D positional encoding fails to preserve 4D coordinates. Another line of work decomposes 4D space into static geometry and scene flow (Wu et al., 2025). While modeling motion in 4D, this representation fails to handle general dynamic scenes involving object appearance and disappearance.

To overcome these constraints, we propose a novel 4D VAE in native 4D space that preserves explicit spatiotemporal coordinates while compressing complete 4D content. We adopt dynamic color voxels as a unified representation to capture both 4D appearance and motion. This choice enables generalization across arbitrary dynamic scenes, including scene content changing, and can naturally represent both

complete (Figure 1 (c), (d)) and partial (Figure 1 (a), (b)) 4D. Crucially, every latent token in our framework retains a 4D coordinate in native 4D space, eliminating projection artifacts and enabling direct spatial-temporal manipulation.

One challenge of building a native 4D VAE is designing a suitable network for native 4D data processing. Modern convolutional operators (Paszke et al., 2019) do not natively support 4D convolution for voxel encoding, unlike in video VAEs (Wan et al., 2025). A naive workaround is to decompose a 4D convolution into separate 3D spatial and 1D temporal convolutions. However, this approach processes spatial and temporal information in isolated stages, resulting in indirect spatio-temporal modeling. To enable more natural interaction between spatial and temporal features, we propose a transformer-based spatio-temporal module (STD/STU) that divides the voxel sequence into local 4D windows and performs attention-based encoding and decoding. This design also allows our VAE to support flexible temporal compression ratios, enabling joint training across sequences with varying frame counts.

Another challenge of training native 4D presentations is suitable training loss for 4D data. Supervision based solely on common losses such as BCE (Goodfellow et al., 2016), DICE (Milletari et al., 2016), or MSE (Ruppert, 2004) often leads to overly smooth reconstructions. To mitigate this, we introduce a novel voxel rendering loss that significantly improves both geometric and color reconstruction quality.

The last challenge of 4D VAE is data format unification. Since our VAE operates directly in native 4D voxel space, all 4D data can be represented as dynamic voxels for training. However, directly converting raw 4D data, such as depth videos, into sequences of dynamic voxels often results in spatial discontinuities that can hinder network convergence. To address this, we propose a new preprocessing pipeline that first converts input data into a depth-based mesh sequence, from which continuous and consistent voxel sequences are generated.

Experiments on 4D VAE reconstruction tasks show that our proposed VAE outperforms existing point map-based VAEs and flow-based VAEs (Wu et al., 2025) in reconstruction quality and temporal stability. Our model not only reduces artifacts but also learns a more direct and structurally consistent latent space for 4D content. Furthermore, we demonstrate the generative potential of our representation by training video-conditioned 4D diffusion models on Hot3D (Banerjee et al., 2025) dataset. These results confirm that our 4D VAE can serve as an efficient backbone for high-fidelity 4D generative modeling.

## 2. Related Work

Recent progress in dynamic scene reconstruction (Wang et al., 2025a) and 4D generation increasingly leverages

strong image/video priors to predict geometry in the form of 2D-aligned maps or point-based representations. Geo4D (Jiang et al., 2025b) and GeometryCrafter (Xu et al., 2025) repurpose large video generators to infer dynamic geometry by predicting multiple complementary geometric maps (e.g., point/disparity/ray-maps) from monocular videos, demonstrating strong generalization from synthetic training to real data. In parallel, feed-forward 4D generation frameworks such as 4DNeX generate dynamic point clouds from a single image by adapting pretrained video diffusion models and modeling spatiotemporal RGB/XYZ trajectories (Chen et al., 2025b). While effective for view-conditioned or partially observed settings, these paradigms naturally inherit view dependence and typically compress 4D structure into 2D-arranged tokens, which makes them less suitable as a holistic 4D compression backbone when native spatiotemporal coordinates must be preserved. More discussion of related work can be found in the appendix.

## 3. Methodology

Our 4D VAE is designed as follow. Given a raw 4D input (e.g., depth video or inherent 4D content), the first stage is to convert it into a sequence of dynamic colored voxels: $V = \{v_i \in \mathbb{R}^{N \times C}\}_{i=1}^{M}$, where $C = 6$ denotes voxel attributes (3 coordinates and 3 color channels), $N$ is the voxel count of each frame, and $M$ is the total number of frames. Details of this stage will be described in Sec. 3.1.

The next stage is the actual latent feature learning using VAE. The encoder part $\mathcal{E}$ is trained to perform spatio-temporal compression, yielding latent tokens $L = \{l_i \in \mathbb{R}^{N' \times C'}\}_{i=1}^{M'}$, where $N' \ll N$ is the token count and $M' < M$ is the compressed number of frames (typically $M' = M/4$, following video VAE settings). A decoder $\mathcal{D}$ then reconstructs high-resolution voxel sequences from these latents $L$. To support native 4D space VAE, we design a novel network structured, described in Sec. 3.2 and Sec. 3.3. Also, we design a novel loss function to enhance visual quality, described in Sec. 3.4 and Sec. 3.4.2. We now describe each component in detail.

### 3.1. Preprocessing

To accommodate heterogeneous 4D data sources—such as single-view RGB-D video, multi-view aggregated point cloud sequences, or dynamic mesh sequences—we unify them into a common representation: dynamic colored voxel sequences.

A straightforward pipeline is to extract 4D point clouds from raw data and voxelize them. For example, depth videos can be back-projected into 3D point clouds using camera intrinsics and extrinsics, followed by voxelization. However, these back-projected point clouds often exhibit spatial discontinuities (e.g., gaps between adjacent points), which persist after voxelization, as shown in Figure 3 (a). Since

our method explicitly models spatio-temporal coherence, such inconsistencies hinder accurate occupancy inference in continuous 4D volumes.

To address this, we introduce an additional preprocessing step that first converts raw data into 4D mesh sequences Figure 3 (b). Specifically, we adopt the Depth-to-Mesh (Hu et al.) method, which connects neighboring back-projected points into mesh faces and applies filtering to produce clean mesh sequences. These mesh sequences are then voxelized, with color values blended within each voxel, resulting in a consistent and high-quality dynamic voxel representation.

### 3.2. Native Spatio-Temporal Window Attention Module

One way to encode dynamic 3D data is to mimic the process used by video VAEs for handling video sequences. Standard video VAEs rely heavily on 3D convolution operators for encoding $h \times w \times t$ patches. However, the naive extension to 4D convolution over $(x, y, z, t)$ is not supported in common frameworks (e.g., PyTorch (Paszke et al., 2019)). Instead of adopting the naive solution to decouple 4D convolution into cascaded blocks of 3D convolution and 1D convolution, we leverage transformers to encode tokens directly in native 4D space. The use of attention for 4D modeling enables efficient information aggregation in 4D and supports flexible temporal compression ratio. Specifically, we propose a Spatio-Temporal Downsampling (STD) Block to compress tokens and use a Spatio-Temporal Upsampling (STU) Block to decode tokens in both spatial and temporal dimensions.

**Spatio-Temporal Downsampling (STD) Block.** Given a voxel token sequence $S = \{s_i \in \mathbb{R}^{N \times C}\}_{i=1}^{M}$, where each $s_i$ carries per-voxel 4D coordinates $p_i \in \mathbb{R}^{N \times 4}$ and $M$ is the number of frames, our goal is to compress $S$ both temporally and spatially into $S' = \{s'_j \in \mathbb{R}^{N' \times C}\}_{j=1}^{\lfloor M/\lambda \rfloor}$, with temporal ratio $\lambda \in \mathbb{N}$ and spatial ratio $\gamma > 1$ (so that $N' \approx N/\gamma^3$ for uniform grids).

*Spatial downsampling.* We first obtain spatially downsampled tokens via sparse convolution:

$$\hat{s}_i = \text{SparseConv}(s_i; \text{factor} = \gamma) \in \mathbb{R}^{N' \times C}, \quad (1)$$

where $i = 1, \ldots, M$.

*4D positional encoding.* We add frequency-based 4D positional encodings (extended from the 3D variant in (Wan et al., 2025)) to both resolutions:

$$s_i \leftarrow s_i + \text{PosEnc}(p_i), \ \hat{s}_i \leftarrow \hat{s}_i + \text{PosEnc}(\gamma \hat{x}_i, \gamma \hat{y}_i, \gamma \hat{z}_i, \hat{t}_i), \quad (2)$$

where $p_i = (x_i, y_i, z_i, t_i) \in \mathbb{R}^{N' \times 4}$ are the coordinates associated with $\hat{s}_i$ and the spatial scaling $\gamma$ aligns the coarse tokens with the $\lambda$-frame groups defined below.

*Windowing and cross-scale attention.* To avoid the prohibitive cost of global 4D attention, we partition the

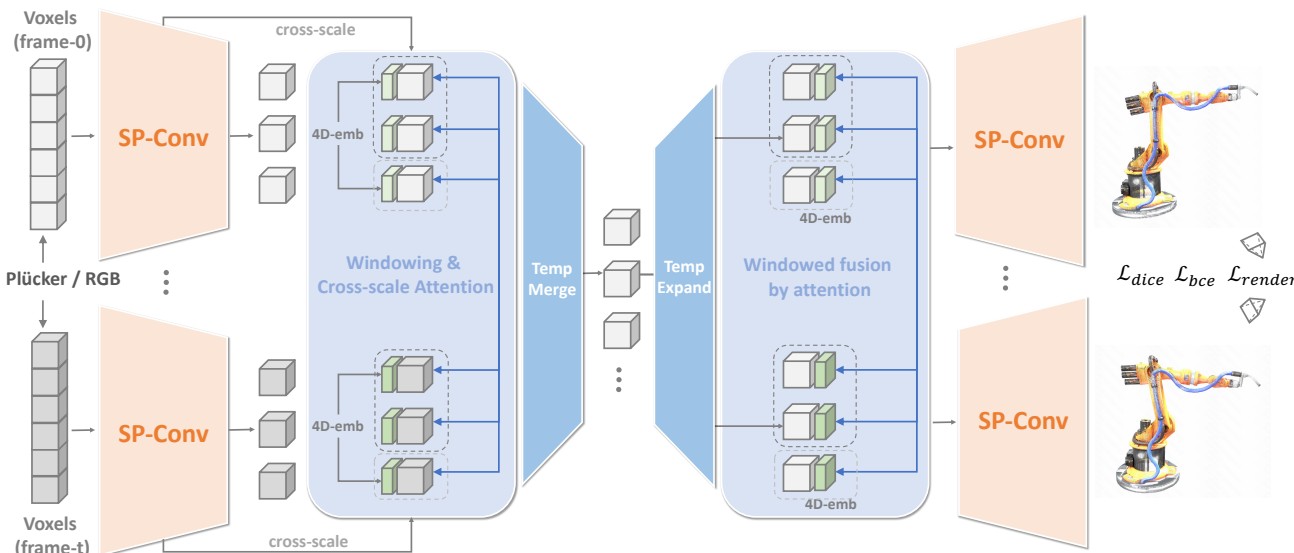

*Figure 2.* Overall framework of our native spatio-temporal 4D VAE.

$(x, y, z, t)$ domain into local 4D windows. Let $\mathcal{W}_\omega(\cdot)$ select tokens that fall into spatial window $\omega$. For the $j$-th group

$$\mathcal{G}_j = \{ i \mid i \in [\lambda(j-1)+1, \ \lambda j] \}, \qquad (3)$$

we aggregate information by querying from the coarse tokens and attending to the union of fine and coarse tokens within each window:

$$\underbrace{Q_{j,\omega} = \mathcal{W}_\omega\Big( \{\, \hat{s}_i \,\}_{i \in \mathcal{G}_j} \Big),}_{\text{queries}}$$
$$\underbrace{K_{j,\omega}, V_{j,\omega} = \mathcal{W}_\omega\Big( \{\, s_i \,\}_{i \in \mathcal{G}_j} \cup \{\, \hat{s}_i \,\}_{i \in \mathcal{G}_j} \Big),}_{\text{keys/values}} \qquad (4)$$

$$\tilde{s}'_{j,\omega} = \mathrm{MHA}\big(Q_{j,\omega}, K_{j,\omega}, V_{j,\omega}\big) + \mathrm{FFN}\big(Q_{j,\omega}\big), \quad (5)$$

and we collect window outputs, average tokens within the same spatial position but different temporal index to form

$$s'_j = \mathrm{Merge}\big(\{\tilde{s}'_{j,\omega}\}\big) \in \mathbb{R}^{N' \times C}. \qquad (6)$$

Intuitively, Eq. (4) lets each coarse token $\hat{s}_i$ (queries) absorb temporally local and spatially neighboring information from both resolutions, yielding compressed tokens $S'$ that preserve native 4D structure.

**Spatio-Temporal Upsampling (STU) Block.** Given $S = \{\, \hat{s}_i \in \mathbb{R}^{N \times C} \,\}_{i=1}^{M}$, STU symmetrically upsamples tokens to $S' = \{\, s'_j \in \mathbb{R}^{N' \times C} \,\}_{j=1}^{\lambda M}$, with the same temporal factor $\lambda$ and a spatial upsampling ratio $\gamma$.

*Temporal expansion with 4D PE.* Each $\hat{s}_i$ is expanded to $\lambda$ finer time steps. Let $\hat{p}_i = (\hat{x}_i, \hat{y}_i, \hat{z}_i, \hat{t}_i)$ be its coordinates; we generate

$$\big[\, \hat{s}_i^{(k)} \,\big]_{k=0}^{\lambda-1} = \big[\, \hat{s}_i + \mathrm{PosEnc}\big(\hat{x}_i, \hat{y}_i, \hat{z}_i, \lambda \hat{t}_i + k\big) \,\big]_{k=0}^{\lambda-1}. \qquad (7)$$

*Windowed fusion by attention.* For each target fine frame index $j$ and window $\omega$, we build queries from the temporally expanded tokens and keys/values from the union of expanded tokens:

$$Q_{j,\omega} = \mathcal{W}_\omega\big(\{\hat{s}_i^{(k)}\}_{(i,k) \mapsto j}\big),$$
$$K_{j,\omega}, V_{j,\omega} = \mathcal{W}_\omega\big(\{\hat{s}_i^{(k)}\}_{(i,k) \mapsto j} \cup \{s_i\}\big), \qquad (8)$$

where $(i, k) \mapsto j$ denotes the mapping from the coarse index $i$ and sub-frame $k$ to the fine frame $j = i\lambda + k$. Windowed attention and merging give

$$\hat{s}_j = \mathrm{Merge}\Big(\big\{\mathrm{MHA}(Q_{j,\omega}, K_{j,\omega}, V_{j,\omega}) + \mathrm{FFN}(Q_{j,\omega})\big\}_\omega\Big) \qquad (9)$$

*Spatial upsampling.* We then spatially lift tokens by sparse up-convolution:

$$s'_j = \mathrm{SparseUpConv}\big(\hat{s}_j; \text{ factor} = \gamma\big) \in \mathbb{R}^{N' \times C}. \quad (10)$$

### 3.3. Architecture

We instantiate the VAE with the above STD/STU blocks. For the *encoder*, given a dynamic colored voxel sequence $V = \{v_i \in \mathbb{R}^{N \times C}\}_{i=1}^{M}$, we first concatenate camera Plücker embeddings per voxel:

$$v_i \leftarrow \big[\, v_i, \ \mathrm{Plucker}(v_i - o_i, \ o_i) \,\big], \qquad (11)$$

where $o_i$ is the camera origin of the $i$-th frame. Camera information is injected only in the encoder. We then apply several spatial-only sparse 3D convolution downsampling blocks to obtain

$$s_i = \mathrm{SparseConv}(v_i) \in \mathbb{R}^{\tilde{N} \times D}. \qquad (12)$$

Next, a stack of STE blocks performs native spatio-temporal compression:

$$s'_j = \text{STD}(\{s_i\}_{i \in \mathcal{G}_j}; \lambda, \gamma) \in \mathbb{R}^{\bar{N}' \times D}, \quad (13)$$

where $j = 1, \ldots, \lfloor M/\lambda \rfloor$. Finally, we densify $s'_j$ and pass them through Conv3D and MLP layers to obtain the latent tokens

$$L = \{l_j = \text{MLP}(\text{Conv3D}(\text{dense}(s'_j)))\}_{j=1}^{\lfloor M/\lambda \rfloor}. \quad (14)$$

For the *decoder*, we mirror the process. STU blocks first upsample the latent features in space and time:

$$\tilde{s}_i = \text{STU}(l_{\lceil i/\lambda \rceil}) \in \mathbb{R}^{\bar{N} \times D}, \quad i = 1, \ldots, M. \quad (15)$$

An MLP predicts per-token occupancy, which we sparsify using a threshold $\tau = -0.5$:

$$\mathcal{S} = \{\tilde{s}_i \mid \sigma(\text{MLP}_{\text{occ}}(\tilde{s}_i)) > \tau\}. \quad (16)$$

Finally, stacked sparse up-convolutions and MLP heads regress color and occupancy:

$$V_{\text{color}} = \left\{ \tanh(\text{MLP}_{\text{rgb}}(\text{SparseUpConv}(\tilde{s}_i))) \right\}_{i=1}^{M},$$
$$V_{\text{occ}} = \left\{ \sigma(\text{MLP}_{\text{occ}}(\text{SparseUpConv}(\tilde{s}_i))) \right\}_{i=1}^{M} \quad (17)$$

where $\sigma$ and $\tanh$ is sigmoid and tanh activation function.

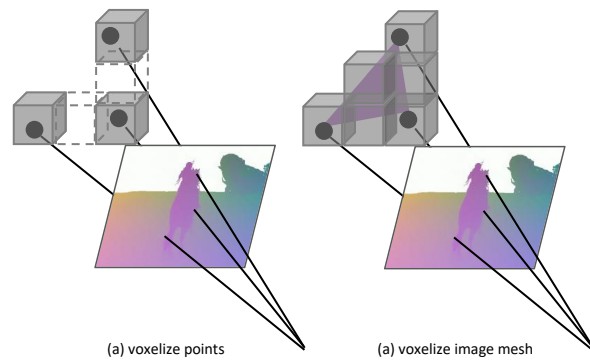

(a) voxelize points          (a) voxelize image mesh

*Figure 3.* Illustration of our mesh-based voxelization.

### 3.4. Optimization loss

#### 3.4.1. 4D LOSS

Once obtain the reconstructed dynamic voxel sequence, we can directly compute losses in 4D space, including BCE loss, DICE loss and RGB MSE loss. Binary Cross-Entropy (BCE) loss $\mathcal{L}_{\text{bce}}$ supervises the error between predicted occupancy and ground truth:

$$-\frac{1}{MN} \sum_{i=1}^{M} \sum_{j=1}^{N} \left[ y_{i,j} \log(v_{\text{occ}}^{i,j}) + (1 - y_{i,j}) \log(1 - v_{\text{occ}}^{i,j}) \right], \quad (18)$$

where $y$ is the ground truth occupancy label, $M$ is frame number and $N$ is voxel number of each frame.

DICE loss can be computed as follows:

$$\mathcal{L}_{\text{dice}} = 1 - \frac{2 \sum_{i=1}^{M} \sum_{j=1}^{N} v_{\text{occ}}^{i,j} y_{i,j} + \epsilon}{\sum_{i=1}^{M} \sum_{j=1}^{N} v_{\text{occ}}^{i,j} + \sum_{i=1}^{M} \sum_{j=1}^{N} y_{i,j} + \epsilon}, \quad (19)$$

where $\epsilon = 1e - 5$. In addition, we also compute the intermediate DICE and BCE loss for early pruning in Equation (16).

RGB MSE loss computes the error of the predicted RGB of each voxel:

$$\mathcal{L}_{\text{rgb}} = -\frac{1}{M \times N} \sum_{i=1}^{M} \sum_{j=1}^{N} \|v_{\text{rgb}}^{i,j} - y_{\text{rgb}}^{i,j}\|_2^2, \quad (20)$$

where $y_{\text{rgb}}$ is the ground truth color.

#### 3.4.2. VOXEL RENDERING LOSS

Although direct supervision in 4D voxel space is effective, we observe that predicted colors can become over-smoothed and sub-optimal. To alleviate this, we introduce a differentiable voxel rendering loss. Differentiable voxel rendering is non-trivial: ray-based sparse-voxel methods (e.g., Plenoxel (Fridovich-Keil et al., 2022)) rely on coarse-to-fine sparsification and are difficult to integrate into feed-forward training, while mesh-extraction approaches (e.g., FlexiCubes (Shen et al., 2023)) convert voxels to meshes, which loses detail and scales poorly to high-resolution grids. We therefore extend Sparse Voxel Rasterization (Sun et al., 2025) from an overfitting regime to a batched, feed-forward setting that supports high-resolution voxel grids.

Given a reconstructed dynamic voxel sequence $V$, we treat predicted occupancy as opacity and predicted color as albedo, and render supervision images via

$$\hat{I}_{\text{rgb}}^i = \text{SparseVoxelRast}(v_{\text{rgb}}^i, v_{\text{occ}}^i).$$

With a per-pixel validity mask $\mathcal{M}^i \in \{0, 1\}^{H \times W}$, the rendering loss is

$$\mathcal{L}_{\text{render}} = \frac{1}{M} \sum_{i=1}^{M} \| \mathcal{M}^i \odot (\hat{I}_{\text{rgb}}^i - I_{\text{rgb}}^i) \|_2^2. \quad (21)$$

Our final objective combines voxel-space and rendering supervision:

$$\mathcal{L} = \beta_1 \mathcal{L}_{\text{bce}} + \beta_2 \mathcal{L}_{\text{dice}} + \beta_3 \mathcal{L}_{\text{rgb}} + \beta_4 \mathcal{L}_{\text{render}} + \beta_5 \mathcal{L}_{\text{kl}}, \quad (22)$$

where $\beta_1 = \beta_2 = \beta_4 = 1$, $\beta_3 = 0.5$, and $\beta_5 = 0.001$ are loss weight.

## 4. Experiments

This section presents the experimental evaluation of our proposed method. Our 4D VAE is trained on a diverse collection of datasets including TartanAir (Wang et al., 2020),

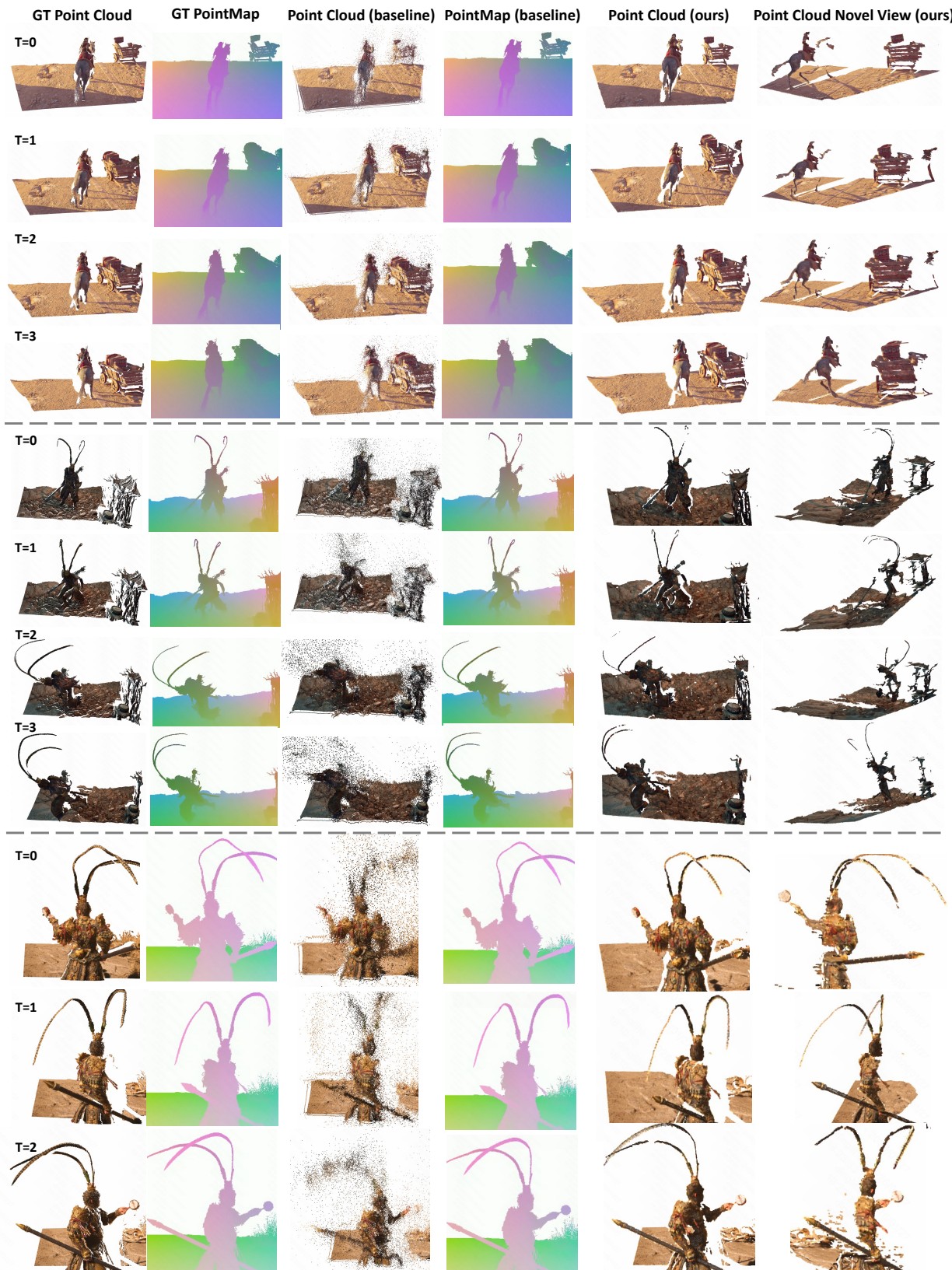

*Figure 4.* Qualitative comparison with PointMap VAE.

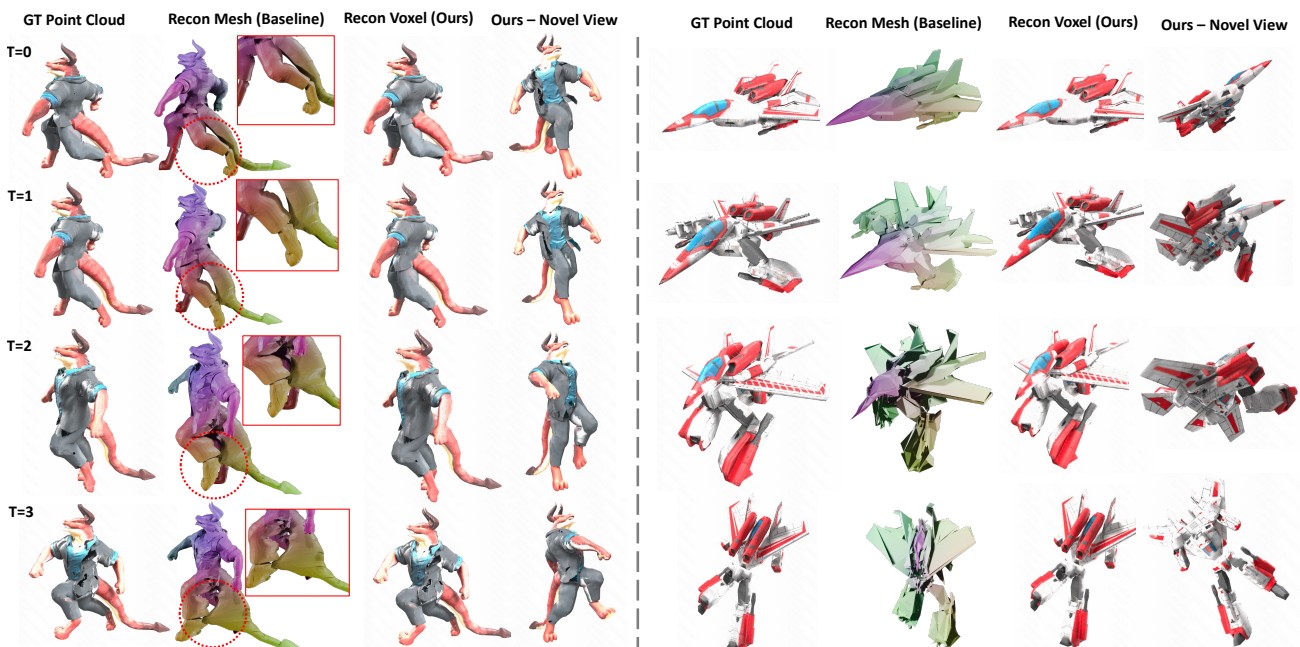

*Figure 5.* Qualitative comparison with Flow-based VAE.

MVS-Synth (Huang et al., 2018), PointOdyssey (Zheng et al., 2023), SynScapes (Wrenninge & Unger, 2018), Spring (Mehl et al., 2023), Dynamic Replica (Karaev et al., 2023), Co3D (Reizenstein et al., 2021), Hot3D (Banerjee et al., 2025), Objaverse (Deitke et al., 2023), DexYCB (Chao et al., 2021), Wild-RGBD (Xia et al., 2024), and OmniWorld (Zhou et al., 2025). We follow (Wang et al., 2025b) to preprocess training datasets and construct a test set named 4D-Dyn, comprising 140 sequences from the test split of these datasets. To further demonstrate the generative applicability of our VAE, we train a video-conditioned 4D hand-object reconstruction model on Hot3D (Banerjee et al., 2025). More details can be found in the appendix.

### 4.1. Implementation Details

Our 4D VAE is trained for 7 days on 32 NVIDIA A800 GPUs using the AdamW optimizer ($\beta_1 = 0.9$, $\beta_2 = 0.999$), with a batch size of 1 and a learning rate of $1e-5$. During training, the input voxel sequences have a spatial resolution of $1024 \times 1024 \times 1024$ and various temporal lengths from 1 to 32 frames. The temporal and spatial compression ratios $\lambda$ and $\gamma$ are set to [1, 4] and 16, respectively. We apply a valid voxel mask during the computation of the rendering loss to exclude invalid voxels identified during preprocessing. More details can be found in the appendix.

### 4.2. Results

#### 4.2.1. QUALITATIVE COMPARISON WITH POINTMAP

Previous pointmap diffusion methods (Jiang et al., 2025b; Xu et al., 2025) either depend on heavy post-optimization for pointmap generation or do not open-source the encoder, we compare our native 4D VAE against the direct video

VAE baseline (Wan et al., 2025). As shown in Figure 4, Despite the fact that video VAE reconstructs pointmaps with visually imperceptible errors, mapping each point back into the 3D space based on its xyz value reveals noticeable discrepancies. While our approach achieves superior geometric fidelity and structural coherence. Moreover, benefiting from the native 4D representation, our VAE does not exhibit distortion artifacts under novel viewpoints. Figure 1 also show some visulization of our reconstructed 4D contents.

#### 4.2.2. QUALITATIVE COMPARISON WITH FLOW-BASED

We also compare our method with flow-based 4D VAE DyMeshVAE (Wu et al., 2025), which encodes decoupled static mesh and vertices flows. Figure 5 shows that DyMesh-VAE fails to robustly handle large motions and complex object transformations, while our VAE achieves high-quality, complete 4D autoencoding. Moreover, DyMeshVAE can not encode colors into its latent space. In contrast, our VAE naturally fuses geometric and RGB information.

#### 4.2.3. VIDEO-CONDITIONED GENERATIVE MODEL

To validate the suitability of our VAE's latent space for generative tasks, we train a 4D latent diffusion model using the latent tokens produced by our encoder. We adopt a transformer architecture based on WAN-VACE (Jiang et al., 2025a), which conditions a DiT backbone on an input video. Our VAE's latent tokens replace the standard latent representations, enabling video-conditioned 4D generation. As illustrated in Figure 6, the diffusion model successfully produces high-quality 4D reconstructions from input videos, confirming the expressiveness and generative utility of our learned latent space. Unlike pointmap reconstruction methods, which only predict partial point clouds, our model is

*Table 1.* Quantitative comparison on 4D-Dyn. Pre means Precision.

| Method | Pre@0.005 ↑ | Recall@0.005 ↑ | Pre@0.001 ↑ | Recall@0.001 ↑ | CD-L1 ↓ | PSNR ↑ |
|---|---|---|---|---|---|---|
| PointMapVAE | 0.4721 | 0.5494 | 0.0387 | 0.0395 | 0.024 | - |
| **Ours** w/o ST-Attn | 0.8723 | 0.9090 | 0.6507 | 0.7812 | 0.0025 | 27.63 |
| **Ours** w/o render loss | 0.9892 | 0.9969 | 0.8853 | 0.8942 | 0.0011 | 25.08 |
| **Ours** w/o Plücker | 0.9976 | 0.9991 | 0.9552 | 0.9835 | 0.0004 | 29.07 |
| **Ours** Full | **0.9981** | **0.9997** | **0.9876** | **0.9918** | **0.0003** | **29.63** |

*Figure 6.* Complete 4D hand-object reconstruction from video.

capable of reconstructing complete 4D hands and objects. More results Figure 9 and details of the video-conditioned diffusion model can be found in the appendix.

### 4.2.4. QUANTITATIVE EVALUATION

Table 1 and Table 2 reports quantitative results on the 4D-Dyn and DyMesh. For 4D-Dyn, we apply a depth threshold to mitigate noise in distant regions of certain samples. We evaluate the following metrics: (i) *Geometric Precision* @threshold $\tau$, (ii) *Geometric Recall* @threshold $\tau$ (iii) *Chamfer Distance-L1*, and (iv) *PSNR* for color reconstruction. Our method consistently outperforms all baselines.

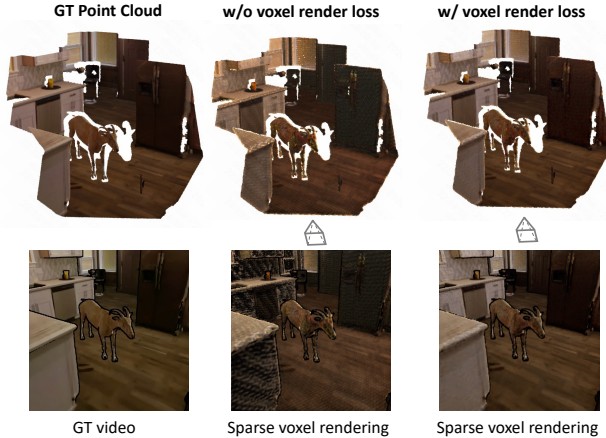

*Figure 7.* Voxel rendering loss ablation results.

### 4.3. Ablation Studies

**Native Spatio-Temporal Attention Block.** To evaluate the effectiveness of our proposed native spatio-temporal attention module, we replace it with a decoupled structure consisting of 3D spatial and 1D temporal convolutions. The second and fourth row in Table 1 show that our native spatio-temporal attention module significantly improves reconstruction fidelity.

**Voxel Rendering Ablation.** We further assess the impact of the voxel rendering loss. As illustrated in Fig. 7, the inclusion of this loss leads to better color reconstructions, whereas its absence results in color artifacts. This observation is also supported quantitatively in Table 1, where the same performance trend is evident.

**Plücker Embedding Ablation.** Table 1 demonstrates that incorporating Plücker embeddings enhances the encoder's ability to utilize camera information, thereby improving rendering quality. Notably, the Plücker embedding is only introduced in the encoder—the decoder operates without any camera input. This design allows the latent tokens to be used in generative modeling without requiring camera parameters during decoding.

**Backbone Analysis.** To strengthen the PointMapVAE baseline, we introduce a normal loss to supervise the normal maps of the reconstructed point clouds against the ground truth, and further fine-tune it on PointMap data with the same number of training iterations as our method. We

*Table 2.* Quantitative comparison on DyMesh dataset. Pre means precision.

| Method | Pre@0.001 ↑ | Recall@0.001 ↑ | Pre@0.0005 ↑ | Recall@0.0005 ↑ | Chamfer-Distance-L1 ↓ |
|---|---|---|---|---|---|
| DyMesh VAE | 0.9945 | 0.9894 | 0.7972 | 0.9254 | 0.0008 |
| **Ours** | **0.9985** | **0.9996** | **0.8895** | **0.9729** | **0.0001** |

*Table 3.* Backbone analysis. Pre means precision.

| Method | Pre@0.001 ↑ | Recall@0.001 ↑ | Pre@0.0005 ↑ | Recall@0.0005 ↑ | CD-L1 ↓ |
|---|---|---|---|---|---|
| PointMapVAE (normal-loss) | 0.5380 | 0.6521 | 0.0577 | 0.0593 | 0.019 |
| PointMapVAE (vggt) | 0.7425 | 0.8390 | 0.2401 | 0.2566 | 0.008 |
| **Ours** | **0.9981** | **0.9997** | **0.9876** | **0.9918** | **0.0003** |

*Table 4.* Mesh pre-process analysis. Pre means precision.

| Method | Pre@0.001 ↑ | Recall@0.001 ↑ | Pre@0.0005 ↑ | Recall@0.0005 ↑ | CD-L1 ↓ |
|---|---|---|---|---|---|
| Ours w/o mesh-pre | 0.9812 | 0.9903 | 0.8579 | 0.8803 | 0.0014 |
| **Ours** | **0.9981** | **0.9997** | **0.9876** | **0.9918** | **0.0003** |

further replace the CNN backbone in the PointMapVAE baseline with a pure transformer architecture following VGGT (Wang et al., 2025a). Specifically, we reuse VGGT's backbone (frame and global attention) as the encoder and reuse its DPT head as the decoder. As shown in Table 3, although the transformer-based architecture improves performance over the convolution-based baseline, our method still surpasses the pure-transformer PointMapVAE, demonstrating the superiority of our native 4D representation.

**Mesh pre-process analysis.** Raw point clouds are discontinuous in 3D space, and directly voxelizing them introduces high-frequency holes. These holes cause abrupt, irregular transitions between occupied and unoccupied regions in the ground truth, which can confuse the network. Therefore, we convert pointmaps to mesh to provide smooth occupancy supervision. We further ablate this design in Table 4. Additionally, we identify edges between different layers based on depth values and apply a threshold to remove stretched meshes between separated layers, which helps avoid most mesh artifacts. For other failure modes, some synthetic data contain smoke or floaters that introduce artifacts when converted to meshes; such data segments were excluded from the dataset.

## 5. Conclusion

In this paper, we propose a novel spatio-temporal VAE operating directly in native 4D space – dynamic colored voxel sequences. Experiments show that our VAE outperforms pointmap-based VAEs and flow-based VAEs for both partial and complete 4D content autoencoding. We also demonstrate the generative potential of our VAE latent by training video-conditioned 4D diffusion models.

## Impact Statement

This paper presents work whose goal is to advance the field of Machine Learning. There are many potential societal consequences of our work, none which we feel must be specifically highlighted here.

## Acknowledgement

This study was supported by CUHK-CUHK(SZ)-GDSTC Joint Collaboration Fund No. 2025A0505000053

RGC Early Career Scheme (ECS) No. 24209224

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

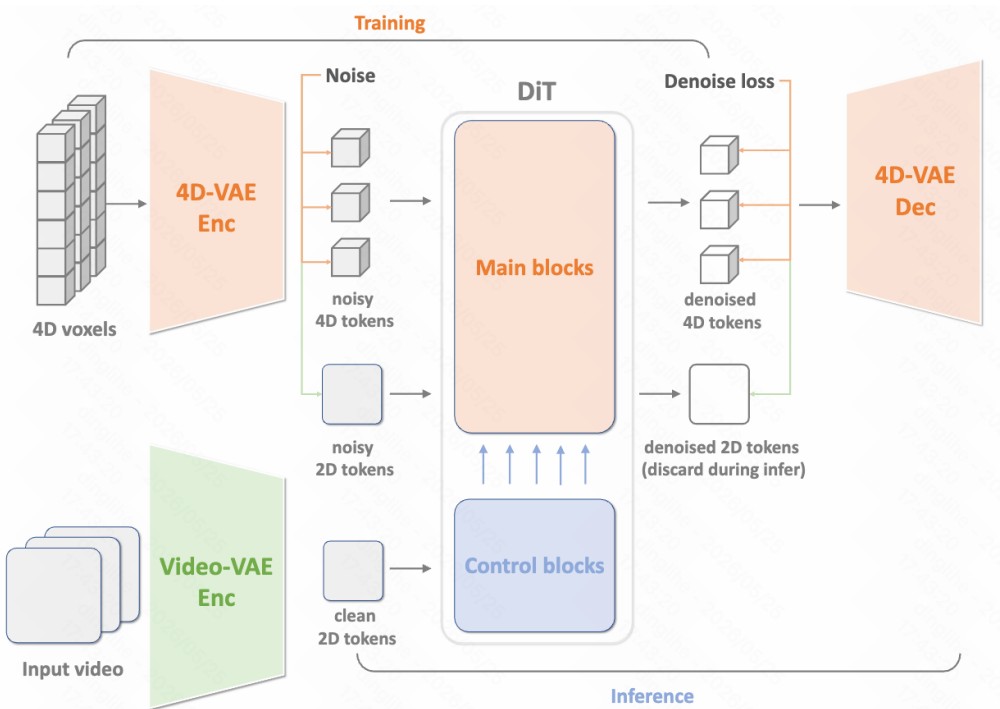

*Figure 8.* Training and inference pipeline of our video-conditioned 4D Diffusion model.

## A. More details of Generative Model

To validate the generative capability of our latent space, we train a video-conditioned 4D diffusion model based on the 1.3B DiT architecture from WAN-VACE (Jiang et al., 2025a). The original VACE model is a video-conditioned video generation model designed for editing, inpainting, and image-to-video tasks. Specifically, the clean conditional video tokens are processed through control blocks and added to the noisy 2D tokens. We retain the video-condition injection module (the blue block in Figure 8) to incorporate input video information. To enable 4D generation, during training, we concatenate the 4D tokens encoded by our VAE with the noisy 2D tokens and compute both 4D and 2D denoising losses. The 2D denoising loss helps the model retain information from the conditional videos. Our 4D tokens have a resolution of $64 \times 64 \times 64$. To reduce memory overhead during diffusion training, we follow the video diffusion model (Wan et al., 2025) and patchify the 4D latent tokens to $32 \times 32 \times 32$. The 4D diffusion model is trained on Hot3D (Banerjee et al., 2025) for complete hand–object reconstruction. We initialize the weights from the pre-trained VACE 1.3B model and fine-tune it on 64 A800 GPUs for 7 days. During inference, we start from pure 4D and 2D Gaussian noise along with clean conditional video tokens, iteratively denoise the 4D and 2D noisy tokens, and finally decode the 4D voxels using the denoised 4D tokens. The denoised 2D tokens are discarded, as they only serve to retain the conditional video information. We also compare our complete reconstruction approach with the point-map-based partial reconstruction method MoGe2 (Wang et al., 2025c).

## B. More Implementation details

Since our spatio-temporal module is flexible, it can accommodate an arbitrary temporal ratio ranging from 1 to 4 across different training iterations, thereby enhancing data efficiency. For instance, while the SynScapes (Wrenninge & Unger, 2018) dataset lacks temporal consistency, we can set the temporal ratio to 1 and utilize it for pre-training. Our training pipeline follows a cascaded strategy: we first fix the temporal ratio to 1 for all datasets to encourage the network to learn spatial compression, then gradually increase the temporal ratio every 4k steps. To reduce memory consumption and improve training efficiency, we adopt an early pruning strategy at a resolution of 256, as described in Equation (16). Ground truth voxels are also processed at this resolution to supervise intermediate occupancy predictions. For data processing, we filter out severely corrupted samples from both training and evaluation data after our depth-to-mesh conversion pipeline, thereby constructing the final training and test sets.

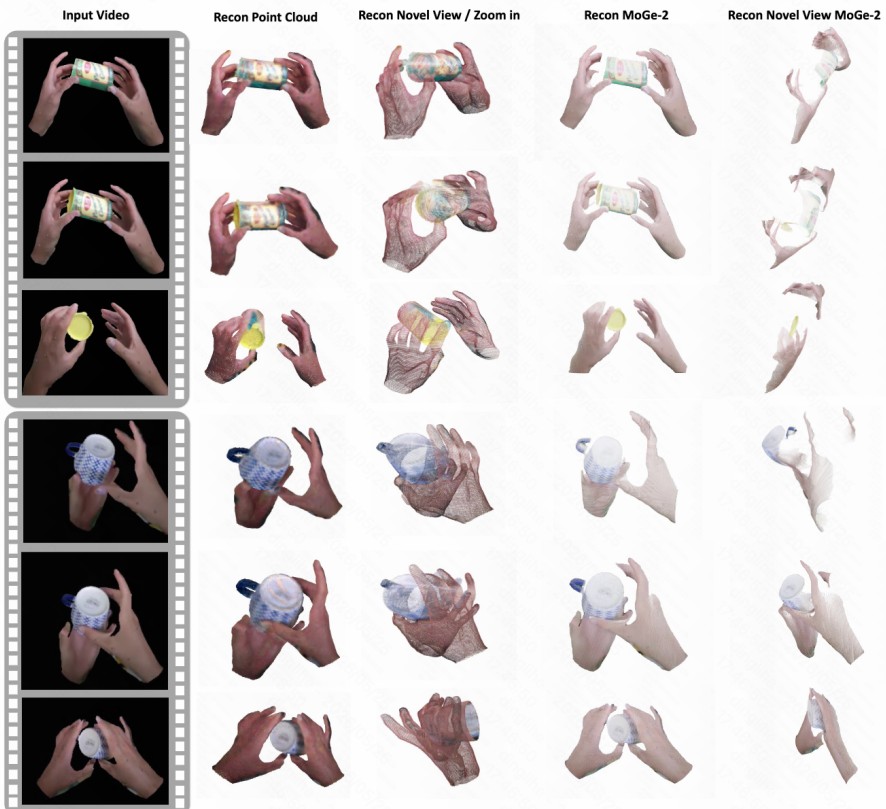

*Figure 9.* More results of video-conditioned video diffusion model.

## C. More related works

**Classical multi-view reconstruction.** Traditional 3D reconstruction is grounded in multi-view geometry, notably Structure-from-Motion (SfM) (Schonberger & Frahm, 2016) and Multi-View Stereo (MVS) (Furukawa et al., 2015). Systems such as COLMAP (Schonberger & Frahm, 2016) follow an incremental pipeline that estimates camera poses and sparse geometry, then refines them via global optimization. These methods explicitly enforce geometric consistency and yield interpretable reconstructions, but they are often computationally heavy and can be brittle in unconstrained settings.

**Feedforward 3D reconstruction.** Recent work has shifted toward end-to-end, feedforward models that infer 3D structure directly from images. DUSt3R (Wang et al., 2024) showed that Transformer-based architectures can reconstruct geometry from unposed, uncalibrated image pairs, and VGGT (Wang et al., 2025a) extended this paradigm to multiple views with global attention. Follow-up methods further explored extensions to dynamic videos (Chen et al., 2025a; Wang et al., 2025b;d). Despite strong progress, many feedforward systems rely on multi-head decoders or multi-model pipelines for different outputs (e.g., depth, pose, point clouds) (Wang et al., 2025b; Zhang et al., 2024; Wang et al., 2025e). Such designs can be cumbersome to run and typically do not provide reliable correspondences in dynamic regions.

## D. Limitations

One cost of directly encoding 4D space is the increased GPU memory consumption and larger number of tokens required compared to pixel-space latent representations. Meanwhile, while mesh-to-voxel preprocessing eliminates discontinuities in dynamic voxels, it may also filter out certain fragmented meshes, such as grass and small rocks. In addition, the performance of our VAE degrades in large-scale scenes due to insufficient voxel resolution.

