# OpenReview forum: "Native Spatio-Temporal 4D Variational Autoencoder"
_ICML.cc/2026/Conference — ICML 2026 regular_

### Official Review · Reviewer_MvmZ · 2026-03-10

**Soundness:** 2
**Presentation:** 3
**Significance:** 2
**Originality:** 3
**Overall Recommendation:** 4
**Confidence:** 3

**Summary:**

This paper proposes a native spatio-temporal 4D VAE for dynamic 3D content. Instead of representing dynamic data through projected 2D point maps, it encodes colored voxels directly in native 4D space and introduces STD/STU blocks for spatio-temporal down/up-sampling, together with a voxel rendering loss to improve geometry and color reconstruction. The paper also shows a video-conditioned 4D generation demo built on the learned latent space.

**Compliance With Llm Reviewing Policy:**

Affirmed.

**Final Justification:**

I admire the authors' efforts for the detailed rebuttal, and my main concerns are addressed. I decide to raise my rating to positive.

**Key Questions For Authors:**

1. Can the authors evaluate all main baselines on both 4D-Dyn and DyMesh, so that the comparison is more complete and directly comparable?
2. Can the authors provide a stronger point-map baseline, or otherwise justify more convincingly why the current PointMapVAE setup is the fairest available comparison? A stronger result here would increase my confidence in the empirical claim.

**Limitations:**

No. The paper should discuss the practical cost of the method more explicitly, and it should also acknowledge the dependence on mesh-based preprocessing and its possible failure modes.

**Strengths And Weaknesses:**

**Strengths**

- The paper addresses a significant problem. A native 4D latent representation is indeed missing in the current literature, and this direction is potentially important for both reconstruction and generation.
- The voxel-based formulation is reasonable and technically promising. Encoding dynamic geometry and color directly in 4D space is a meaningful alternative to projection-based point-map representations.
- The proposed architectural components are sensible. In particular, the STD/STU design is a reasonable way to model joint spatial-temporal structure without native 4D convolutions.
- The voxel rendering loss is a useful implementation contribution and appears to improve reconstruction quality.

**Weaknesses**

- The experimental comparison is incomplete. On 4D-Dyn, the paper mainly compares against PointMapVAE, while on DyMesh it compares against DyMeshVAE; the paper does not evaluate all relevant baselines on both datasets. This makes the empirical picture less convincing.
- The PointMapVAE comparison is not fully fair. The baseline is adapted by using a video-VAE-style design to handle point-cloud data, and although the paper provides a rationale, this setup seems disadvantaged a priori. As a result, the claimed advantage over point-map-based 4D VAEs is not yet fully validated.
- The paper includes a video-conditioned 4D generation demo, but the evidence there is qualitative only. Since generative utility is one of the motivations, quantitative comparisons would substantially strengthen the paper.
- Efficiency is under-reported. The paper states training takes 7 days on 32 A800 GPUs, but does not provide memory, runtime, or inference-cost analysis. Given the heavy voxel representation, these practical costs are important.
- The mesh-based preprocessing is potentially important, but its effect is not isolated. The paper first converts point clouds into mesh sequences before voxelization, yet there is no ablation for this step. More importantly, directly meshing discontinuous point clouds may create stretched connections between separated layers (e.g., foreground/background), and the paper does not discuss this possible failure mode.

---

> ### Author Rebuttal · Authors · 2026-03-31
>
> Thank you for your helpful comments. Our replies to your questions are stated below.
>
>
> **W1 & Q1**: Evaluate all baselines on both 4D-Dyn and DyMesh datasets.
>
> A1:  The 4D-Dyn dataset contains only partial observations (e.g., depth-unprojected points), whereas DyMesh provides complete 4D mesh sequences. The PointMapVAE baseline is designed to handle partial observations defined as pointmaps in image space, while DyMeshVAE operates on mesh vertices and faces and cannot process partial observations. Therefore, we evaluate each baseline with our method on the respective datasets for which they are suited. Additionally, we render the DyMesh mesh sequences into image space to obtain pointmaps, enabling a comparison with PointMapVAE on this rendered partial pointmap data. The table below shows that our method also significantly outperforms PointMapVAE on the rendered DyMesh data.
>
> | **Method** | **Pre@0.005 ↑** | **Recall@0.005 ↑** | **Pre@0.001 ↑** | **Recall@0.001 ↑** | **CD-L1 ↓** |
> | --- | --- | --- | --- | --- | --- |
> | PointMapVAE | 0.5422 | 0.6038 | 0.0875 | 0.0945 | 0.013 |
> | **Ours** | **0.9979** | **0.9991** | **0.8712** | **0.9690** | **0.0002** |
>
>
> **W2 & Q2**: Stronger point-map baseline.
>
> A1: To strengthen the PointMapVAE baseline, we introduce a normal loss to supervise the normal maps of the reconstructed point clouds against the ground truth, and further fine-tune it on PointMap data with the same number of training iterations as our method. Additionally, following GeoCrafter, we add a residual encoder to enhance PointMapVAE and replace the SVD-based autoencoders with the more powerful WAN2.1 VAE. The results in the table below show that while these modifications improve the baseline, our method still outperforms all enhanced baselines, demonstrating the superiority of native 4D modeling.
> | **Method** | **Pre@0.005 ↑** | **Recall@0.005 ↑** | **Pre@0.001 ↑** | **Recall@0.001 ↑** | **CD-L1 ↓** |
> | --- | --- | --- | --- | --- | --- |
> | PointMapVAE (normal) | 0.5380 | 0.6521 | 0.0577 | 0.0593 | 0.019 |
> | PointMapVAE (GeoCrafter) | 0.6746 | 0.7005 | 0.1356 | 0.1763 | 0.010 |
> | **Ours** | **0.9981** | **0.9997** | **0.9876** | **0.9918** | **0.0003** |
>
>
> **W3**: Quantitative results of generative models.
>
> **A3**: We quantitatively compare our 4D generative model with state-of-the-art methods, including HaMeR and HaWoR, on the Hot3D dataset in the table below. Specifically, we fit MANO hand parameters from the generated voxels and compare the results with the ground truth in world space. We report Procrustes-Aligned Mean Per Joint Position Error (PA-MPJPE), World MPJPE (W-MPJPE), and World-Aligned MPJPE (WA-MPJPE). The results demonstrate the superiority of generative modeling in native 4D space.
>
> | **Method** | **PA-MPJPE ↓** | **W-MPJPE ↓** | **WA-MPJPE ↓** |
> | --- | --- | --- | --- |
> | HaMeR | 9.39 | 156.03 | 43.37 |
> | HaWoR | 4.79 | 33.20 | 11.27 |
> | **Ours** | **3.44** | **28.54** | **10.62** |
>
>
> **W4**: Computational cost and efficiency analysis.
>
> **A4**: We report the latency of mesh pre-processing, voxelization (res=1024), and model forward pass in the table below, along with the memory cost of model inference. The results are averaged over 100 standard temporal input chunks, where each chunk contains 4 frames of pointmaps and each frame contains approximately 260k points.
>
> | **Stage** | **Latency (s)** | **GPU Memory (GB)** |
> | --- | --- | --- |
> | mesh preprocess | 1.956 | - |
> | voxelization | 1.279 | - |
> | encoding | 0.2266 | 11.3 |
> | decoding | 0.5960 | 16.7 |
>
>
> **W5**: Mesh pre-process analysis.
>
> **A5**: Raw point clouds are discontinuous in 3D space, and directly voxelizing them introduces high-frequency holes. These holes cause abrupt, irregular transitions between occupied and unoccupied regions in the ground truth, which can confuse the network. Therefore, we convert pointmaps to mesh to provide smooth occupancy supervision. We further ablate this design in the table below. Additionally, we identify edges between different layers based on depth values and apply a threshold to remove stretched meshes between separated layers, which helps avoid most mesh artifacts. For other failure modes, some synthetic data contain smoke or floaters that introduce artifacts when converted to meshes; such data segments were excluded from the dataset.
>
> | **Method** | **Pre@0.005 ↑** | **Recall@0.005 ↑** | **Pre@0.001 ↑** | **Recall@0.001 ↑** | **CD-L1 ↓** |
> | --- | --- | --- | --- | --- | --- |
> | Ours w/o mesh-pre | 0.9812 | 0.9903 | 0.8579 | 0.8803 | 0.0014 |
> | Ours | 0.9981 | 0.9997 | 0.9876 | 0.9918 | 0.0003 |

---

> > ### Author Rebuttal · Reviewer_MvmZ · 2026-04-03
> >
> > I thank the authors for their detailed rebuttal. The addition of quantitative metrics for the generative model, the efficiency analysis, and the mesh pre-processing ablation are appreciated and effectively address my concerns in those specific areas.
> >
> > However, my core concern regarding the Point-map baseline (W2 & Q2) remains unresolved. While I acknowledge the authors' effort to strengthen the baseline by incorporating a normal loss and GeoCrafter-like modifications, I argue that this is moving in the wrong direction.
> >
> > The fundamental issue lies in the base architecture. The current baseline relies on a CNN-based Video-VAE. The inductive biases of Convolutional Neural Networks (e.g., translation invariance and local receptive fields) are explicitly designed for 2D image domains. They are fundamentally ill-suited for point-map data, where the channel dimensions encode explicit 3D spatial coordinates (XYZ) rather than just color/intensity. Applying auxiliary losses or residual connections to an incompatible CNN backbone is structurally sub-optimal and does not reflect the true potential of point-map representations.
> >
> > A genuinely fair and credible point-map baseline should abandon the CNN backbone in favor of a pure Transformer architecture, drawing inspiration from recent feed-forward point-map reconstruction models like DUSt3R or VGGT (which are already contextualized in the literature).
> >
> > Because the PointMap baseline is inherently limited by an unsuitable architecture, the claim that the proposed native 4D representation is strictly superior to point-map representations remains unconvincing.

---

> > > ### Author Response · Authors · 2026-04-06
> > >
> > > Thank you for the supportive assessment and the constructive suggestion. We further replace the CNN backbone in the PointMapVAE baseline with a pure transformer architecture following VGGT. Specifically, we reuse VGGT's backbone (frame and global attention) as the encoder and reuse its DPT head as the decoder.
> > >
> > > Moreover, instead of using 3D convolutions, we achieve 4× temporal compression through attention. The input frames are first encoded by a ViT and then grouped into non-overlapping 4-frame chunks.. We replace the camera token in VGGT's backbone with a temporal register token that has two learnable variants: one for the first frame of each chunk and another for the remaining three frames. These temporal register tokens are prepended to the patch tokens of the corresponding frames before entering the backbone. After the alternating frame-attention and global-attention layers, we obtain an aggregated feature list as in VGGT. We then select tokens from layer {4, 11, 17, 23} following VGGT, and for each 4-frame chunk, retain only the first frame's patch tokens—which have already absorbed information from the other three frames via global attention—thereby achieving 4× temporal compression along the time axis. The selected four layers' tokens are concatenated along the channel dimension and projected via an MLP to a 256-channel spatial latent, preserving the patch-grid spatial structure.
> > >
> > > For decoding, the per-chunk latent tokens are first repeated to represent every frame within the chunk, concatenated with 4 temporal register tokens, and processed through a global-attention module to recover frame-specific information. Note that temporal and spatial RoPE embeddings are also applied during attention. The resulting per-frame tokens are then passed through four separate MLPs to produce four feature inputs that are fed into the DPT head. The DPT head upsamples these features and regresses a point map and a confidence map, following the same procedure as in VGGT.
> > >
> > > We adopt the same loss functions as VGGT for pointmap supervision: a confidence-weighted L2 regression loss and a surface normal loss. In addition, we apply a KL divergence regularization term with β = 1×10⁻⁴. We initialize the corresponding weights of the attention and DPT layers from VGGT, and fine-tune the transformer-based PointMapVAE baseline for the same number of training iterations as our method. As shown in the table below, although the transformer-based architecture improves performance over the convolution-based baseline, our method still surpasses the pure-transformer PointMapVAE, demonstrating the superiority of our native 4D representation.
> > >
> > > | **Method** | **Pre@0.005 ↑** | **Recall@0.005 ↑** | **Pre@0.001 ↑** | **Recall@0.001 ↑** | **CD-L1 ↓** |
> > > | --- | --- | --- | --- | --- | --- |
> > > | PointMapVAE (normal) | 0.5380 | 0.6521 | 0.0577 | 0.0593 | 0.019 |
> > > | PointMapVAE (VGGT) | 0.7425 | 0.8390 | 0.2401 | 0.2566 | 0.008 |
> > > | **Ours** | **0.9981** | **0.9997** | **0.9876** | **0.9918** | **0.0003** |
> > >
> > > We will include this discussion and experiment in the revised manuscript. Thank you again for your constructive suggestion.

---

### Official Review · Reviewer_jSeK · 2026-03-11

**Soundness:** 4
**Presentation:** 3
**Significance:** 3
**Originality:** 4
**Overall Recommendation:** 5
**Confidence:** 3

**Summary:**

This paper introduces a novel native 4D Variational Autoencoder (VAE) to address the limitations of existing 4D VAEs, which typically rely on 2D pointmap projections or scene flow and consequently suffer from projection-induced artifacts and difficulties with topological changes. To overcome these issues, the authors propose representing dynamic scenes using dynamic color voxels, thereby explicitly preserving spatiotemporal coordinates in native 4D space. To process this data effectively, the framework features a transformer-based spatio-temporal module that applies attention mechanisms across local 4D windows, enabling direct and joint spatial-temporal modeling rather than disconnected 3D+1D processing. Furthermore, the method introduces a novel voxel rendering loss to prevent over-smoothed outputs and a specialized preprocessing pipeline to generate continuous voxel sequences for stable training. Experimental results demonstrate that the proposed VAE significantly outperforms existing pointmap-based and flow-based methods in both reconstruction quality and temporal stability. Finally, the authors validate the framework's effectiveness as a latent backbone for generative tasks by successfully training video-conditioned 4D diffusion models on the Hot3D dataset.

**Compliance With Llm Reviewing Policy:**

Affirmed.

**Final Justification:**

I have decided to raise my final score to 5 as my concerns has been well addressed. I think this paper is in good quality and shows a promising direction towards 4D native VAE for many downstream tasks.

**Key Questions For Authors:**

Please address my concerns mentioned in the weakness section.

**Limitations:**

yes

**Strengths And Weaknesses:**

### Strengths

1. Solid and Novel Contribution: This is a solid and innovative work. The paper proposes a framework that efficiently compresses 4D content into sequences and demonstrates the strong potential of using the pre-trained encoder for video generation. To achieve this, a novel STD block is introduced to efficiently process 4D voxel sequences. Extensive experiments demonstrate promising and compelling results.
2. Clarity and Reproducibility: The paper is overall well-written, and the methodology provides sufficient detail to suggest good reproducibility. It was an enjoyable read.

### Major Weaknesses

1. Unclear Loss Function Motivation: The motivation behind the chosen combination of loss functions is confusing. It is unclear why BCE, DICE, and RGB losses are all strictly necessary. What would happen if only the RGB and rendering losses were used? An ablation study or further discussion on this would be helpful.
2. Comparison with Existing Video Generation Methods: Although direct comparisons with state-of-the-art baselines (e.g., Seedance 2.0) may be difficult, it would be valuable to know whether the proposed 4D encoder offers distinct advantages in specific aspects, such as view consistency or memory efficiency. A thorough discussion comparing this approach with existing video generation methods would significantly strengthen the paper's conclusions and better clarify its positioning in the field.
3. Exposition and Formatting Issues: The exposition of the paper could be improved. The authors should carefully double-check the ordering of the figures, ensure every figure is properly referenced in the text, and verify the consistency of mathematical notations.
4. Reproducibility Concerns: The paper introduces significant engineering efforts and model improvements, making the work challenging to reproduce or evaluate independently. Given the authors' own critique that "previous pointmap diffusion methods either depend on heavy post-optimization... or do not open-source the encoder," will the authors provide pre-trained models and code to ensure the reproducibility of their own work?


### Minor Weaknesses
1. Figure 3 is cited on page 3 and should ideally be placed on the same page. Additionally, there appears to be no in-text citation for Figure 2.
2. The definition of s_i in Section 3.2 is confusing, does it contain color information or just 4D spatial-temporal information {p_i}? In any case, the notation C seems to be abused, and do not align with its definition in Section 3 where C = 6.
3. It would be highly beneficial to provide supplementary generated videos to more vividly demonstrate the visual performance and temporal consistency of the method :)

---

> ### Author Rebuttal · Authors · 2026-03-31
>
> Thank you for your helpful comments. Our replies to your questions are stated below.
>
> **W1**: Loss function Analysis
>
> **A1**: The 4D BCE loss and RGB MSE loss provide direct per-voxel supervision for occupancy and color prediction, playing a key role in stabilizing training. The DICE loss encourages the network to maximize the IoU between the reconstruction and ground truth, helping it handle detailed geometry and escape suboptimal solutions during training. Additionally, supervising color prediction with only MSE loss tends to produce overly smooth colors, so we introduce a rendering loss to encourage the recovery of RGB details. We also ablate the loss designs in the table below. The results show that native 4D losses, including BCE and DICE, are essential for spatial supervision—using only RGB and rendering losses leads to a significant performance drop.
>
> | **Method** | **Pre@0.005 ↑** | **Recall@0.005 ↑** | **Pre@0.001 ↑** | **Recall@0.001 ↑** | **CD-L1 ↓** | **PSNR ↑** |
> | --- | --- | --- | --- | --- | --- | --- |
> | Ours w/o BCE | 0.9915 | 0.9959 | 0.9702 | 0.9834 | 0.0005 | 28.65   |
> | Ours w/o DICE | 0.9897 | 0.9871 | 0.8320 | 0.8477 | 0.0018 |  26.51   |
> | Ours (only RGB & render) | 0.4802 | 0.5541 | 0.0395 | 0.0527 | 0.021 |  25.19  |
> | Ours | 0.9981 | 0.9997 | 0.9876 | 0.9918 | 0.0003 | 29.63 |
>
> **W2**: Discuss with video generative models
>
> **A2**: Video generative models train diffusion models on 2D latents and achieve impressive results by leveraging large-scale video data. Although 4D data at a similar scale is not yet available for native 4D diffusion, the native 4D latent space holds significant potential for **world models**, **long-context memory**, and **4D control**. i) Current world models primarily incorporate action control into video diffusion models to predict future visual frames but do not model future geometric 4D structure. Our 4D VAE provides a native 4D latent space that can be trained jointly with 2D latents to simultaneously model both future visual frames and 4D geometry. ii) Long-context learning requires storing long sequences of 2D latent frames for video diffusion models, whereas our native 4D latents reduce redundant storage for the same scene context, as static content at the same spatial location can be compressed into a single latent. iii) We can encode 4D conditional content using our VAE and inject control information via latent cross-attention with 2D latents. We will explore these directions in future work.
>
> **W3**: Figure ordering and notations.
>
> **A3**: We will reorder the figures and refine the mathematical notations in the revised manuscript.
>
> **W4**: Open-Source.
>
> **A4**: We will open-source our weights and code to support and advance 4D modeling research within the community.
>
> Minor weaknesses
>
> **A1**: We will place Figure 3 on page 3 and add a textual citation for Figure 2 in Section 3 in the revised manuscript.
>
> **A2**: The $s_i$ in Section 3.2 is a SparseTensor that contains both 4D coordinates and color information. The color and spatial information are encoded into a $C$-channel feature. We will refine the notation in the revised manuscript for clarity.
>
> **A3**: We will provide supplementary videos in the revisied manuscript.

---

> > ### Author Rebuttal · Reviewer_jSeK · 2026-04-01
> >
> > The authors' rebuttal successfully addressed my concerns regarding the loss selection strategy and provided a concrete discussion on the video generation model. Additionally, the authors have committed to improving the paper's exposition and releasing their code and model weights, which will enhance the reproducibility of their work. Therefore, my concerns are fully resolved.

---

> > > ### Author Response · Authors · 2026-04-01
> > >
> > > Thank you for your thoughtful assessment and encouraging feedback. We greatly appreciate your constructive suggestions and will carefully revise the manuscript accordingly.

---

### Official Review · Reviewer_McMm · 2026-03-12

**Soundness:** 3
**Presentation:** 3
**Significance:** 3
**Originality:** 3
**Overall Recommendation:** 4
**Confidence:** 4

**Summary:**

This work proposes a native 4D VAE that represents dynamic content as colored voxel sequences rather than 2D pointmap projections. The architecture features spatio-temporal window attention with flexible temporal compression, trained with a differentiable voxel rendering loss. The method outperforms pointmap and flow-based VAEs, and a video-conditioned diffusion model trained on the learned latents demonstrates generation potential.

**Compliance With Llm Reviewing Policy:**

Affirmed.

**Key Questions For Authors:**

1. Please provide detailed architecture specifications: how many layers does the encoder and decoder have? What are the channel dimensions and attention head counts at each scale? What is the total parameter count? This information is essential for assessing reproducibility and efficiency.

2. Can you provide a fairer comparison against dedicated pointmap VAEs, such as the encoder components from Geo4D or GeometryCrafter? The current baseline using a generic video VAE to encode pointmaps appears to be a weak strawman.

3. What are the inference times for encoding and decoding a single 4D sequence? How many active voxels are there per frame? How does your method compare to pointmap VAEs in inference speed?

**Limitations:**

Please refer to the weakness and question section.

**Strengths And Weaknesses:**

# Strengths

- **Well-motivated paradigm shift**. Working directly in 4D voxel space rather than projecting to 2D is a meaningful departure from prior approaches. The work clearly articulates why pointmap-based methods are fundamentally limited, and native 4D representation naturally resolves both issues. The motivation is convincing.

- **Strong quantitative results**. Results on 4D-Dyn (Table 1) are impressive: Precision @0.001 reaches 0.9876 vs. 0.0387 for PointMapVAE, with an ~80× reduction in CD-L1 (0.024 vs. 0.0003). Table 2 confirms consistent gains on DyMesh. The ablation study cleanly isolates each component's contribution that spatio-temporal attention is the most critical, while the voxel rendering loss primarily improves color quality.

- **Validated generative potential**. The video-conditioned 4D diffusion model (Figure 6) demonstrates that the learned latent space supports generative modeling, strengthening the case for this method as a general-purpose 4D backbone beyond reconstruction.

# Weaknesses

- **Potentially unfair baseline comparison**. The manuscript compares against a generic video VAE (Wan et al., 2025) applied to pointmaps; therefore, a representation mismatch that likely explains the extremely poor baseline numbers (Precision @0.001: 0.0387). A fairer comparison would use purpose-built pointmap VAEs such as the encoders from Geo4D or GeometryCrafter.

---

> ### Author Rebuttal · Authors · 2026-03-31
>
> Thank you for your helpful comments. Our replies to your questions are stated below.
>
> **W1 & Q2**: Compare with GeoCrafter and Geo4D.
>
> **A1**: We follow GeoCrafter by adding a residual encoder to enhance the PointMapVAE and replace the SVD-based autoencoders with the more powerful WAN2.1 VAE, then fine-tune it under the same settings as our method. We also compare with the PointMapVAE used in Geo4D without post-optimization. The table below on 4D-Dyn shows that our method significantly outperforms image-space baselines, demonstrating the superiority of native 4D autoencoding.
>
> | **Method** | **Pre@0.005 ↑** | **Recall@0.005 ↑** | **Pre@0.001 ↑** | **Recall@0.001 ↑** | **CD-L1 ↓** |
> | --- | --- | --- | --- | --- | --- |
> | PointMapVAE (Geo4D) | 0.5790 | 0.6133 | 0.0602 | 0.0763 | 0.015 |
> | PointMapVAE (GeoCrafter) | 0.6746 | 0.7005 | 0.1356 | 0.1763 | 0.010 |
> | **Ours** | **0.9981** | **0.9997** | **0.9876** | **0.9918** | **0.0003** |
>
> **Q1**: Network architecture
>
> **A1**: We provide the architecture specifications of the encoder and decoder in the table below. The total parameters of the VAE is **320 MB**. We will also open-source our code and weights to support and advance 4D modeling research within the community.
>
> | **Encoder** |  **num** | **Kernel** | **Channel** | **Head_num** |
> | --- | --- | --- | --- | --- |
> | Linear | 1 | - | (12, 32) | - |
> | Conv3D block | 3 | 3 $\times$ 3 $\times$ 3 | (32, 64, 128, 512) | - |
> | STD block | 2 | - | 512 | 8 |
> | Linear | 2 | - | (512, 512, 32) | - |
>
> | **Decoder** |  **num** | **Kernel** | **Channel** | **Head_num** |
> | --- | --- | --- | --- | --- |
> | Linear | 1 | - | (16, 512) | - |
> | STU block | 2 | - | 512 | 8 |
> | Conv3D block | 3 | 3 $\times$ 3 $\times$ 3 | (512, 128, 64, 32) | - |
> | Linear | 3 | - | (32, 1), (512, 1), (32, 3) | - |
>
>
> **Q3**: Inference latency analysis
>
> **A3**: We report the inference latency comparisons on a single NVIDIA A100 GPU. The results are averaged by testing on 100 standard temporal input chunk, where each chunk contains 4 fame pointmaps and each frame contains about 260k points and 228k activated voxels.
>
> | **Method** | **Encode (ms)** | **Decode (ms)** | **Total (ms)** |
> | --- | --- | --- | --- |
> | PointMapVAE | 303 | 178 | 481 |
> | PointMapVAE (GeoCrafter) | 564 | 275 | 839 |
> | Ours | 227 | 596 | 823 |

---

> > ### Author Rebuttal · Reviewer_McMm · 2026-04-02
> >
> > I appreciate the rebuttal. My concerns have been resolved.

---

> > > ### Author Response · Authors · 2026-04-03
> > >
> > > Thank you for your encouraging feedback. We appreciate your constructive suggestions and will carefully revise the manuscript accordingly.

---

### Official Review · Reviewer_g7mW · 2026-03-12

**Soundness:** 3
**Presentation:** 2
**Significance:** 3
**Originality:** 3
**Overall Recommendation:** 5
**Confidence:** 4

**Summary:**

This paper introduces a novel Native Spatio-temporal 4D Variational Autoencoder (VAE) that directly processes dynamic 3D content in a unified colored voxel space rather than relying on projected 2D pointmaps or decoupled static-and-flow representations. To handle the computational challenges of native 4D processing without resorting to decoupled convolutions, the authors propose a transformer-based Spatio-Temporal Downsampling and Upsampling (STD/STU) module that performs attention-based encoding within local 4D windows. The architecture supports flexible temporal compression ratios and utilizes a preprocessing pipeline to convert raw 3D sequence data into consistent dynamic meshes and continuous voxel sequences. To prevent the over-smoothed outputs common with standard Binary Cross-Entropy (BCE) or Mean Squared Error (MSE) objectives, the authors design a differentiable voxel rendering loss based on sparse voxel rasterization to significantly enhance geometric and color reconstruction fidelity.

Extensive experiments across multiple datasets demonstrate that this native 4D approach outperforms state-of-the-art pointmap and flow-based VAE baselines in reconstruction quality, and the resulting structurally consistent latent space is proven effective for training downstream generative models, such as video-conditioned 4D diffusion.

**Compliance With Llm Reviewing Policy:**

Affirmed.

**Final Justification:**

My concerns are address, I maintain my opinion to accept this paper.

**Key Questions For Authors:**

I do not have additional questions other than the ones in the "Weaknesses" section. I am currently inclined to give a rating of Accept as long as the authors can clarify some of the unclear parts of the paper. I am also interested to hear opinions of other reviewers.

**Limitations:**

Yes, in Supp. Mat.

**Strengths And Weaknesses:**

S1. In an obviously relevant area of research, this paper presents what seems to be a highly effective native 4D representation. The performance gains over existing methods are not simply attributable to scale, but to the clever choice of operating in a unified colored voxel space to eliminate projection distortions and token dislocations. The architecture also benefits from a novel spatio-temporal attention module instead of relying on decoupled convolutions. I would like to hear from other reviewers whether the specific pointmap and flow-based baselines evaluated are fully representative of the current literature, as I know there is much work being done in this field.

S2. The qualitative examples provided in the paper are persuasive and provide useful complementary information to the quantitative results. In particular, Figures 4 and 5 do a great job of highlighting the structural coherence of the proposed method compared to the baselines.

S3. The paper is easy to follow and the core motivation for abandoning 2D projected representations is well-structured.

W1. In terms of the preprocessing pipeline, I don't fundamentally see the argument for why it makes sense to convert the raw data into 4D mesh sequences before voxelization. At the very least, the argument that back-projected point clouds exhibit persistent spatial discontinuities is not rigorous. Furthermore, introducing an intermediate meshing step seems computationally heavy. Please provide a much clearer and extensive explanation of the computational overhead involved here, and theoretically why a simpler point cloud filtering approach would not work.

W2. I am also quite confused by the asymmetric use of Plücker embeddings. The authors mention that camera information is injected only in the encoder so the decoder can operate independently for downstream tasks. However, this is not explained with enough depth to justify the design choice. Please elaborate on how this architectural asymmetry impacts the consistency of the learned latent space.

W3. In terms of the token sparsification, the idea of using a hard threshold of -0.5 is not explained at all. This hinders my understanding of the occupancy prediction process. Please explain this design choice more carefully and provide insight into the model's sensitivity to this specific value.

W4. There are no quantitative experiments to justify the window size design choices for the local 4D attention modules presented in Section 3.2. Please provide ablation studies or similar experimental results detailing how different spatial and temporal window dimensions affect both reconstruction performance and runtime latency.

W5. While I appreciate the results for the downstream generative applications shown in Figure 6, I am confused about what it exactly means to replace the standard latent representations with the VAE latent tokens in the transformer architecture. Please provide more concrete details on this integration process.

W6. Figure 2, while visually appealing, does not in my opinion provide any insight into the cross-scale attention pipeline with any amount of clarity. For starters, the data flow into the windowing blocks seems quite hard to trace, and I am not sure what the exact relationship between the different token resolutions is supposed to show me. Please elaborate on what I am supposed to learn from this figure. In other words, it doesn't seem to stand alone as a reasonable explanation of the pipeline.

---

> ### Author Rebuttal · Authors · 2026-03-31
>
> Thank you for your helpful comments. Our replies to your questions are stated below.
>
> **W1**: Mesh pre-process analysis.
>
> **A1**: Raw point clouds are discontinuous in 3D space, and directly voxelizing them introduces high-frequency holes. These holes cause abrupt, irregular transitions between occupied and unoccupied regions in the ground truth, which can confuse the network. A simple point cloud filtering method can remove floaters but fails to address these holes, whereas converting points to mesh can provide smooth occupancy supervision. We further ablate the mesh pre-processing in the table below and the results show the effectivness of this step. In addition, the average inference time of mesh pre-processing for each 4-frame chunk is 1.956s and consumes no GPU memory, as it is implemented on CPU—making it acceptable for a pre-processing operation.
>
> | **Method** | **Pre@0.005 ↑** | **Recall@0.005 ↑** | **Pre@0.001 ↑** | **Recall@0.001 ↑** | **CD-L1 ↓** |
> | --- | --- | --- | --- | --- | --- |
> | Ours w/o mesh-pre | 0.9812 | 0.9903 | 0.8579 | 0.8803 | 0.0014 |
> | Ours | 0.9981 | 0.9997 | 0.9876 | 0.9918 | 0.0003 |
>
> **W2**: Asymmetric use of Plücker embeddings.
>
> **A2**: We incorporate camera information via Plücker embeddings in the encoder to encourage it to encode this information into the latent representation. During decoding, however, camera information is not assumed to be available, so Plücker embeddings are omitted from the decoder. Instead, the camera information is compressed into the latents and implicitly assists the decoder in achieving better voxel rendering. This allows a latent generated by the diffusion model to be decoded directly into 4D content without requiring camera information at inference time.
>
> **W3**: Hard threshold analysis
>
> **A3**: The threshold τ is an empirically determined value used to sparsify voxels after the STU block. Since the network learns to predict a positive logit for activated voxel, any negative threshold within a certain range is theoretically acceptable. To evaluate the model's sensitivity to this threshold, we tested values in the range of
> −1 to 0 and observed stable performance between
> −0.7 and −0.3.
>
> | **τ** | **Pre@0.001 ↑** | **Recall@0.001 ↑** | **CD-L1 ↓** |
> | --- | --- | --- | --- |
> | -0.9 | 0.9787 | 0.9815 | 0.0005 |
> | -0.7 | 0.9854 | 0.9907 | 0.0003 |
> | -0.5 | 0.9876 | 0.9918 | 0.0003 |
> | -0.3 | 0.9869 | 0.9924 | 0.0003 |
> | -0.1 | 0.9801 | 0.9836 | 0.0004 |
>
> **W4**: Window size ablation
>
> **A4**: We provide ablation results for both performance and inference latency across different local window sizes (wx, wy, wz, wt) in the table below. The results show that
> (4,4,4,4) achieves the best performance.
>
> | **window size (wx, wy, wz, wt)** | **Pre@0.001 ↑** | **Recall@0.001 ↑** | **CD-L1 ↓** | **Latency ↓** |
> | --- | --- | --- | --- | --- |
> | (8, 8, 4, 4) | 0.9780 | 0.9799 | 0.0005 |  785 ms |
> | (8, 8, 8, 4) | 0.9809 | 0.9837 | 0.0005 |  760 ms |
> | (4, 4, 4, 4) | 0.9876 | 0.9918 | 0.0003 | 823 ms |
> | (4, 4, 4, 2) | 0.9774 | 0.9905 | 0.0003 | 951 ms |
> | (2, 2, 2, 4) | 0.9769 | 0.9821 | 0.0004 |  864 ms |
>
> **W5**: Generative intergration detail
>
> **A5**: We train a video-conditioned 4D diffusion model based on the DiT architecture from WAN-VACE. We replace the original video latents with our VAE-encoded 4D latents and compute the denoising loss in this latent space, forming a 4D latent diffusion model. Additionally, we inject conditional video tokens via control blocks to enable video-conditioned 4D generation. Specifically, we apply self-attention between the video tokens from the control blocks and the 4D latents. Figure 8 in supp also provides further architectural details..
>
> **W6**: Confusing figure
>
> **A6**: We will update Figure 2 to better illustrate the data flow within the windowing blocks and provide more detailed explanations of the cross-scale attention mechanism, offering a clearer description of the pipeline in the revised manuscript.

---

> > ### Author Rebuttal · Reviewer_g7mW · 2026-04-04
> >
> > My concerns have been resolved.

---

> > > ### Author Response · Authors · 2026-04-06
> > >
> > > Thank you for your encouraging feedback. We will carefully revise the manuscript accordingly.

---

### Decision · Program_Chairs · 2026-04-30

**Decision:**

Accept (regular)

**Comment:**

This paper introduces a native 4D variational autoencoder that represents dynamic scenes as 4D voxels to avoid the projection-induced artifacts commonly observed in previous point map or flow-based 4D VAEs. It employs a novel transformer-based spatio-temporal attention module that seems to effectively mitigate the artifacts. All reviewers recognize the novelty of the approach as well as the improvement over the baseline. Initial reviews raised concerns about the fairness of baseline comparisons, the missing efficiency analysis, the lack of quantitative generative evaluation, and the insufficient ablation studies. The authors addressed these in the rebuttal with strengthened baselines, efficiency statistics, and additional experiments, and all four reviewers confirmed their concerns were resolved, with final ratings of 5, 5, 4, 4. In the final version, the authors are expected to incorporate the transformer-based pointmap baseline comparison, improve Figure 2, refine notations, disseminate the code and weights, and add supplementary videos.